# Diversity in the intrinsic apoptosis pathway of nematodes

Neil D. Young [1], Tiffany J. Harris[2], Marco Evangelista[2], Sharon Tran[2,3], Merridee A. Wouters [2,6], Tatiana P. Soares da Costa[4], Nadia J. Kershaw[5], Robin B. Gasser [1], Brian J. Smith [4], Erinna F. Lee [2,3,4,7 ✉] & W. Douglas Fairlie [2,3,4,7 ✉]

Early studies of the free-living nematode *C. elegans* informed us how BCL-2-regulated apoptosis in humans is regulated. However, subsequent studies showed *C. elegans* apoptosis has several unique features compared with human apoptosis. To date, there has been no detailed analysis of apoptosis regulators in nematodes other than *C. elegans*. Here, we discovered BCL-2 orthologues in 89 free-living and parasitic nematode taxa representing four evolutionary clades (I, III, IV and V). Unlike in *C. elegans*, 15 species possess multiple (two to five) BCL-2-like proteins, and some do not have any recognisable BCL-2 sequences. Functional studies provided no evidence that BAX/BAK proteins have evolved in nematodes, and structural studies of a BCL-2 protein from the basal clade I revealed it lacks a functionally important feature of the *C. elegans* orthologue. Clade I CED-4/APAF-1 proteins also possess WD40-repeat sequences associated with apoptosome assembly, not present in *C. elegans*, or other nematode taxa studied.

[1] Faculty of Veterinary and Agricultural Sciences, The University of Melbourne, Parkville, VIC, Australia. [2] Olivia Newton-John Cancer Research Institute, Heidelberg, VIC 3084, Australia. [3] School of Cancer Medicine, La Trobe University, Melbourne, VIC 3084, Australia. [4] La Trobe Institute for Molecular Science, La Trobe University, Melbourne, VIC 3086, Australia. [5] The Walter and Eliza Hall Institute of Medical Research, Parkville, VIC 3052, Australia. [6] Present address: ProCan®, Children's Medical Research Institute, Faculty of Medicine and Health, The University of Sydney, Westmead, NSW, Australia. [7] These authors contributed equally: Erinna F. Lee, W. Douglas Fairlie. ✉email: Erinna.Lee@latrobe.edu.au; doug.fairlie@onjcri.org.au

Apoptosis is a form of programmed cell death used by all metazoans to remove unwanted, damaged or dangerous cells. The intrinsic ("mitochondrial") cell death pathway is activated in response to cellular stresses, such as DNA damage or growth factor deprivation, and is regulated by the BCL-2 family of proteins[1]. In mammals, this family consists of opposing pro-survival and pro-apoptotic members which are related to each other by regions of sequence homology, called BCL-2 homology (BH) domains[1]. The pro-survival BCL-2 proteins possess four BH domains (BH1–BH4) and inhibit apoptosis by binding directly to the two sub-classes of pro-apoptotic proteins, BAX/BAK/BOK and BH3-only proteins.

BAX, BAK and BOK also possess BH1–BH4 domains and have a similar three-dimensional structure to the pro-survival proteins, though can change conformation following interaction with particular BH3-only proteins ("activators") and assemble into pore-like structures that permeabilise the outer mitochondrial membrane[2–4]. Mitochondrial outer membrane permeabilisation (MOMP) results in the release of factors, such as cytochrome *c*, from the mitochondrial intermembrane space. Cytochrome *c* facilitates the assembly of the apoptosome, a large multimeric complex consisting of the adapter protein APAF-1 which serves as a scaffold for the activation of proteolytic enzymes, the caspases. Activated caspases cleave vital intracellular substrates, leading to the demise of the cell.

While most studies have focused on the intrinsic apoptotic pathway of mammals, BCL-2 proteins have also been discovered in multiple phyla including the Porifera[5–7], Cnidaria[8–10], Mollusca[11], Platyhelminthes[12,13], Arthropoda[14,15] and Nematoda[16–19]. Interestingly, in a small number of taxa whose intrinsic apoptotic pathway has been extensively characterised, distinct differences have been observed in how they are regulated. For example, while mammals[1], platyhelminths[12,20], cnidaria[8] and molluscs[21] (and probably arthropods[15]) appear to possess at least one BAX/BAK/BOK-like protein, no such or similar molecule has been reported for any nematode. Instead, in the best-characterised, free-living nematode, *Caenorhabditis elegans*, the BCL-2 pro-survival protein, CED-9, regulates apoptosis by directly engaging the pro-apoptotic protein, CED-4 (refs. [17,18,22])—an APAF-1 orthologue that does not require cytochrome *c* for its assembly into the apoptosome[23]. The *C. elegans* apoptosis pathway is also relatively simple, with one pro-survival protein and two BH3-only proteins (EGL-1 and CED-13). By contrast, in humans there are five pro-survival proteins, eight BH3-only proteins and BAX/BAK/BOK proteins.

Here, we describe the identification of BCL-2 proteins in 89 nematode taxa, including socioeconomically important parasites of humans, other animals and plants. We show that the BCL-2-regulated pathway in multiple nematodes is distinct from the classically recognised pathway in *C. elegans*, with clear evidence for expansion of the multi-BH domain BCL-2-type proteins in some species and notable structural differences in both the BCL-2 and the APAF-1/CED-4 homologues.

## Results

### Gain and loss of genes encoding BCL-2 proteins in nematodes.
We used the BCL-2 family Pfam domain that represents more than 1500 BCL-2 homologues to search the inferred proteomes of 89 nematode taxa (Supplementary Table 1) in Wormbase and ENSEMBL databases. BCL-2 protein homologues were identified in 12 free-living and 60 animal-parasitic nematode taxa of evolutionary clades I, III, IV and V (no genomic data are publicly available for any clade II nematode), while none was detected in any of the six species that are endoparasites of plants (clade IV; *Meloidogyne*, *Globodera*, *Ditylenchus*) (Supplementary Tables 1

and 2 and Supplementary Data 1 and 2). Unexpectedly, there was a significant variation in the numbers (one to five) of BCL-2-like proteins within individual taxa, though those with multiple family members were restricted to clades I and III, with five (in *Plectus sambesii*, clade I), four (in *Romanomermis culcivorax*, clade I) and two paralogues in multiple taxa of clade I (*Trichuris* spp.) and clade III (e.g., *Brugia* spp., *Dirofilaria immitis*, *Loa loa*, *Wuchereria bancrofti*, among others) (Supplementary Tables 1 and 2). Sequence variation between paralogues within individual species varied markedly from 22–27% (e.g., *Trichuris* spp.) to 10–42% (e.g., *Romanomermis*) upon pairwise comparison (Supplementary Table 3). This apparent gain of genes encoding BCL-2-like proteins in free-living and animal-parasitic taxa (*n* = 15) and loss in the plant-parasitic taxa studied are discordant with the recognised prototypic *C. elegans* apoptosis pathway with one encoded BCL-2 protein[24].

### Crystal structure of a BCL-2-like protein from *Trichuris suis*.
BCL-2 proteins are defined by the presence of four regions of sequence homology, the BCL-2 homology domains (BH1–BH4)[1]. In CED-9, and indeed for all nematode BCL-2 protein family members we identified, all BH domains are relatively well-conserved, though those of clade I are most similar to those in mammals (Supplementary Table 4).

High-resolution structural analyses of CED-9 (refs. [22,25,26]) revealed an unexpected insertion of four amino acids (relative to mammalian BCL-2 orthologues) at the α4–α5 junction (between the BH3 and BH1 domains) (Fig. 1a) that enables dissociation of CED-4 upon EGL-1 binding to trigger the apoptotic cascade. An alignment of BCL-2 protein sequences representing all nematode taxa studied showed that this region was markedly shorter for clade I compared to other clades (Fig. 2). Thus, due to the functional importance of this region in *C. elegans*, we attempted to determine the crystal structure of both BCL-2 proteins of *T. suis* (designated as *T. suis* A and *T. suis* B for convenience), a representative species of clade I. We were only able to obtain crystals for *T. suis* A and determined its structure to high resolution (1.6 Å) (Supplementary Table 5), revealing a classic BCL-2-fold consisting of eight α-helices (Fig. 1b). Helices α3–α4, which line the BH3 binding-groove, are essentially parallel, similar to the apo-BCL-XL structure[2], though closer together, particularly towards the top of the groove, resulting in a "closed" conformation, as seen in several other BCL-2 structures[2]. The end of α3 is noticeably disordered relative to other apo-BCL-2 structures. The unstructured loop connecting α1–α2, which is significantly shorter in *T. suis* BCL-2 (and CED-9) compared with BCL-XL and BCL-2, was well-defined and packs primarily against α1, although with some contacts with the beginning of α6. Helix α1 is well-resolved throughout its entire length and its very N-terminal portion is in close proximity to α7, a feature not evident for other BCL-2 protein structures. Helix α8 is also extended relative to that of BCL-XL and is visible through essentially all of the juxta-transmembrane region, which is usually not seen in most BCL-2 structures. Notably, however, the short loop at the α4–α5 junction observed in the CED-9 structure was absent (Fig. 1c). As this loop is thought to be critical for CED-4 dissociation, this structure indicates that the regulation of apoptosis in *T. suis* is potentially distinct from *C. elegans*.

### Phylogenetic relationships of BCL-2 proteins.
Multiple sequence alignments were made of the nematode BCL-2 family protein (Fig. 2), as well as these proteins with BCL-2 proteins representing a range of distinct phyla (Fig. 3) and used to construct Bayesian interference (BI) trees.

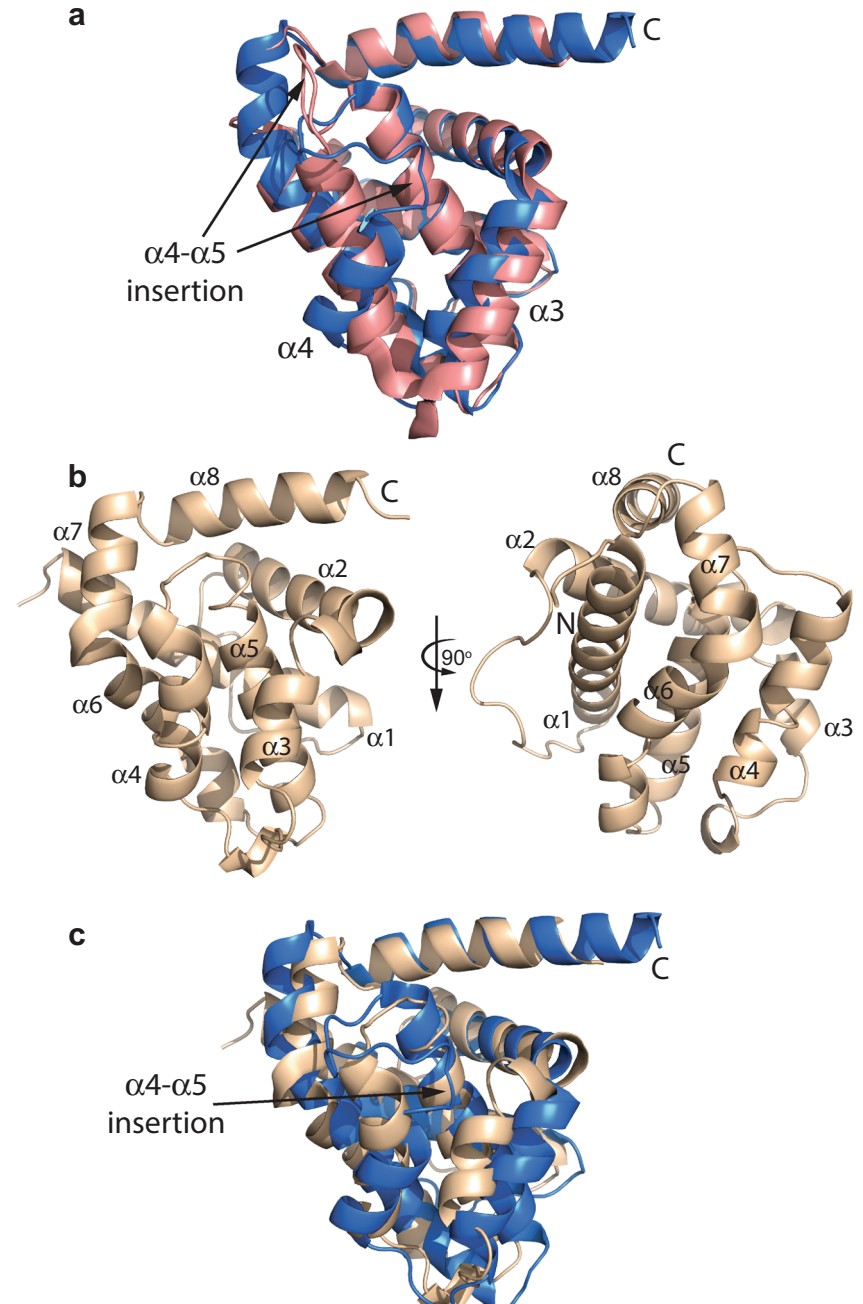

**Fig. 1 Structural analysis of *Trichuris suis* BCL-2. a** Overlay of CED-9 structures. In the structure of unbound CED-9 (PDB ID# 1OHU[25], blue), the loop structure connecting helices α4 and α5 projects into the groove formed by helices α3 and α4. Upon binding EGL-1 (PDB ID# 1TY4 (ref. [26]), pink), the loop undergoes a significant conformational change and is displaced from the groove, coincident with a reorientation and extension of helix α4 (T155-N158) with consequences for the interaction with CED-4. **b** Crystal structure of *T. suis* A. The protein adopts a helical bundle structure similar to other BCL-2 proteins (RMSD 2.4 Å over 141 residues versus BCL-XL (PDB 1PQ0); TM-score: 0.83303 (ref. [69])). Notable features include the entire α1 helix being visible, the well-defined α1–α2 loop and the long α8 helix. **c** The α4–α5 loop in CED-9 (blue) is not present in *T. suis* A (beige).

First, a tree was constructed just using nematode sequence data. The BCL-2 proteins group according to the classes Adenophorea and Secernentea originally proposed by Chitwood and Chitwood[27,28], with most clusters being in accord with the evolutionary clades (I, III-V) (Fig. 2)[29]. For species of clades I and III, with more than one BCL-2 protein, the locations of encoding genes in the genomes suggest that paralogues arose from distinct evolutionary events. In clade III nematodes, the existence of multiple such proteins appears to relate to a recent gene duplication event with both genes being more similar to each

other compared to the clade I paralogues (Supplementary Table 3), and are closely associated with each other on the chromosome. By contrast, the paralogues of clade I representatives (*Trichuris* spp.) are located on distinct genomic scaffolds, and the weaker identity between them is consistent with this clade being basal to the others, and therefore having had a longer time to evolve more specialised functions. Notably, the expansion of BCL-2 proteins in clade III nematodes was restricted to those of the order Spirurida (subclade IIIc) which clustered to the exclusion of those of the order Ascaridida (subclade IIIb). The

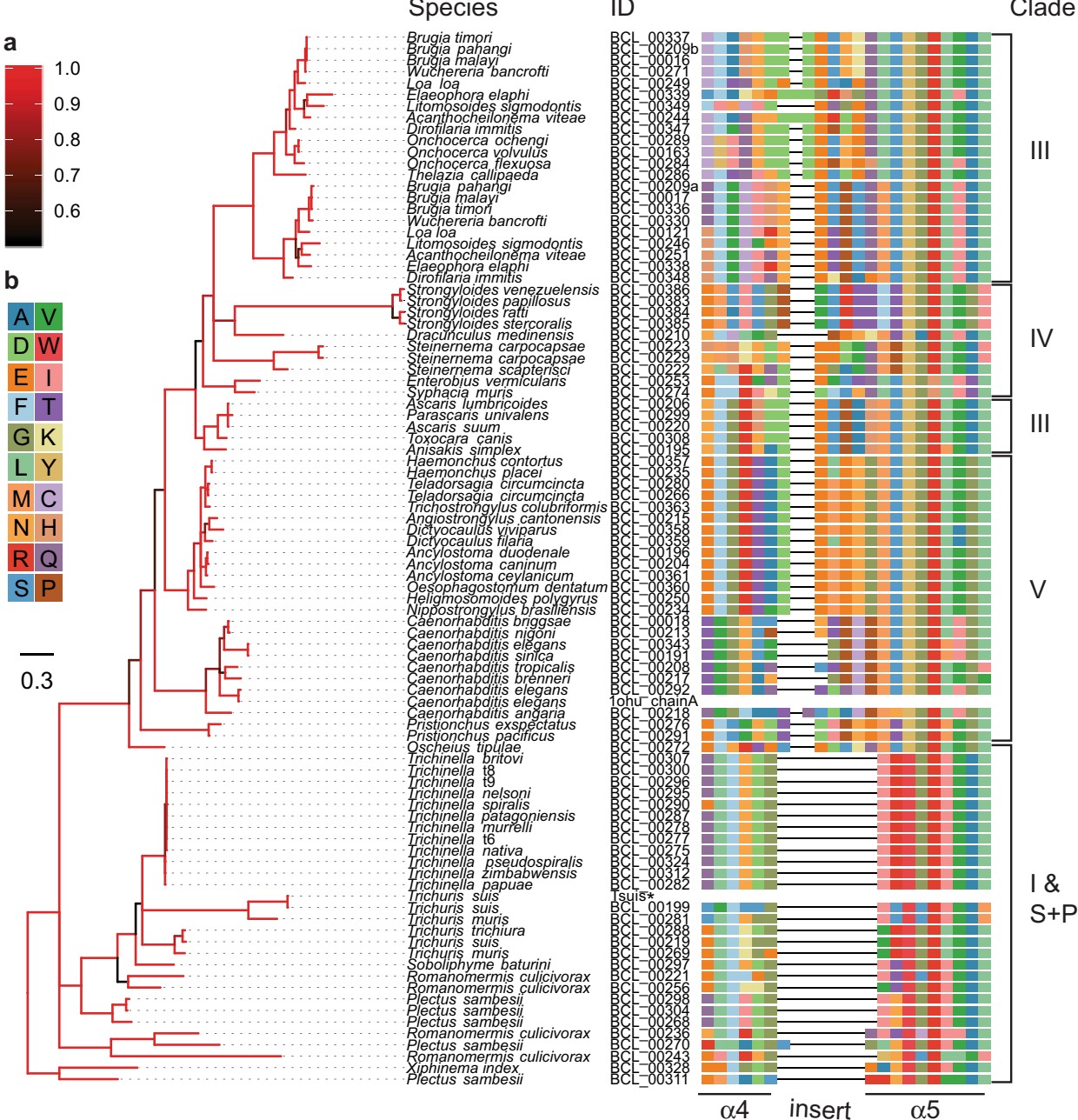

**Fig. 2 Relationships among predicted nematode BCL-2 proteins and alignment of the region between the BH3 on helix α4 and the BH1 domain on helix α5.** Consensus Bayesian phylogenetic tree of predicted nematode BCL2 proteins. Tree tips are labelled by species name and sequence identity number (see Supplementary Table 2). **a** Branches are coloured by the posterior probability support values of adjacent nodes, where black is 0.5 and red is 1.0. **b** Colours of each amino acid residue in the selected region of the Bcl-2 sequence alignment. Tree scale represents 0.3 substitutions per site. *Trichuris suis sequence derived from resolved crystal structure (see Fig. 1) #Caenorhabditis elegans sequence derived from the published three-dimensional structure (PDB ID# 1OHU chain A) with an artificially larger gap between helices α4 and α5 due to there not being any electron density for that part of the structure.

species of clade V were split into two sub-groups with those of *Caenorhabditis* spp. being distinct from the others consistent with the division between the free-living rhabditid and the parasitic strongylid groups, respectively, within the suborder Rhabditina. Finally, the nematode BCL-2 proteins of *Strongyloides* spp. (clade IV) were clearly distinct outliers, with a very long tree branch separating these proteins suggestive of diversification, and perhaps reflecting their complex free-living and parasitic life-cycles with both sexual and asexual reproductive development[30].

Next, a tree was constructed using BCL-2 protein data for nematodes as well as members of other phyla (Fig. 3). This tree is far more complex and its construction was confounded by the likely combination of speciation events and gene expansions giving rise to proteins with distinct pro-survival and pro-apoptotic functions that characterise the multi-BH domain BCL-2 family members. Nodal support was strongest for the proteins from chordate species, which are over-represented compared with other phyla, reflective of the greater availability of genomes for vertebrates. These proteins clustered in accord with the distinct functional groups (i.e. BCL-2/BCL-XL/BCL-W, MCL-1, BFL-1, BOK, BAX, BAK, and some of the lesser-characterised groups such as NR-13, BCL-2L13, -14 and -15).

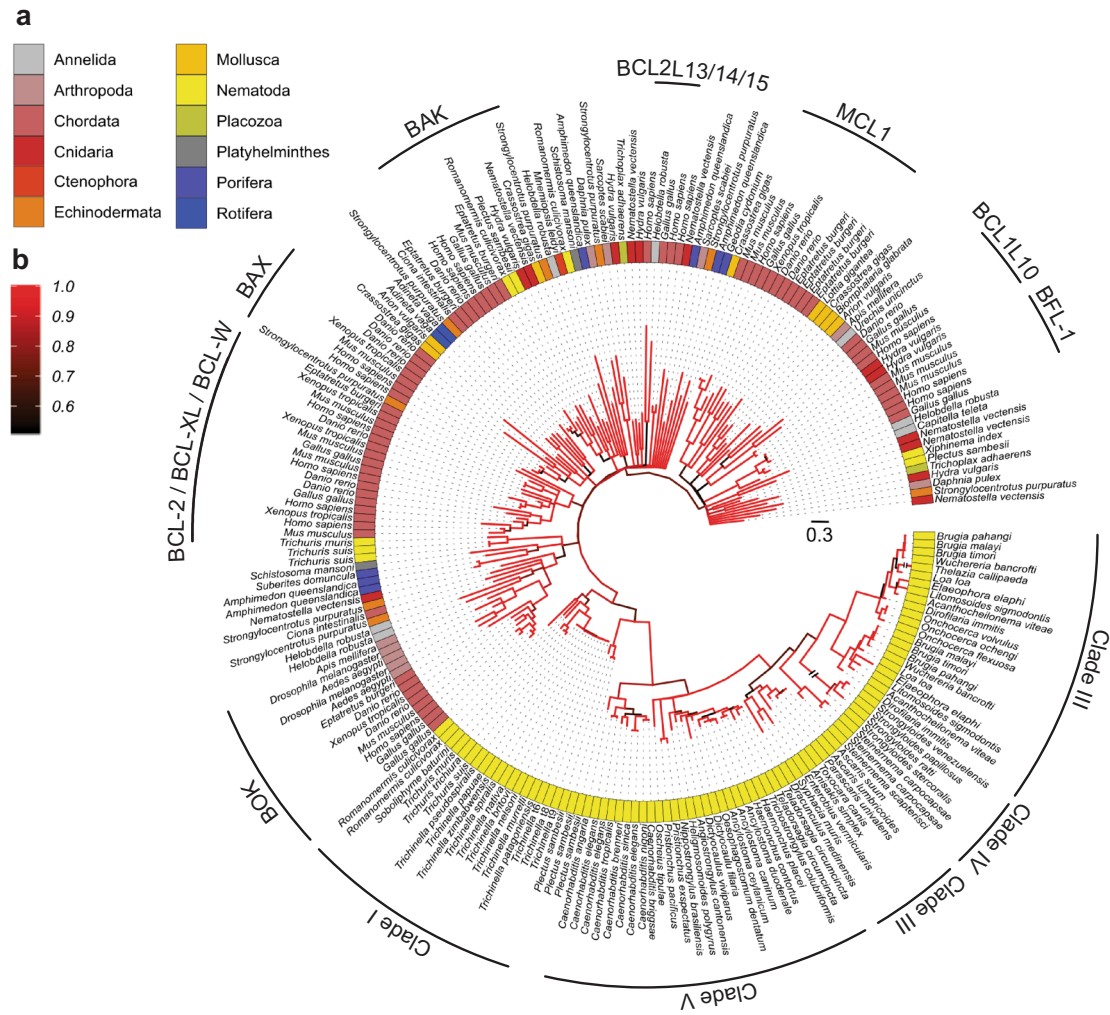

**Fig. 3 Relationships among predicted nematode BCL-2 proteins and other eukaryotic taxa.** Consensus Bayesian phylogenetic tree of predicted nematode BCL-2 proteins and representative eukaryotic taxa. Functionally classified BCL-2 families are labelled. **a** Tree tips are labelled by their taxonomic grouping (phylum) and species name. **b** Branches are coloured by the posterior probability support values of adjacent nodes, where black is 0.5 and red is 1.0. Tree scale represents 0.3 substitutions per site.

From this tree, it was apparent that proteins encoded in the genomes of clade I taxa were basal to other clades, as indicated previously[29,31]. However, in this larger tree, it was evident that one of the two BCL-2 paralogues from *T. suis* and *T. muris* segregated into a distinct cluster with sponges and cnidaria, considered basal. Although this cluster has weak nodal support (posterior probability: 0.73), this topology does suggest that the BCL-2 proteins of these clade I nematodes were under a distinct selection pressure compared with their paralogues as well as those representing other clades (III, IV and V).

Conspicuous in the second tree were two BCL-2 proteins representing clade I nematodes that associated with chordate orthologues, but are likely to have evolved independently from known vertebrate BCL-2 proteins. In particular, one each of the *R. culculovirax* and *P. sambesii* proteins was associated with the BAK proteins. The protein of *R. culculovirax* is unusual in that it has a significantly truncated N-terminus (starting immediately before the BH3 domain but lacking a BH4 domain), suggestive of a structurally distinct protein, while the *P. sambesii* protein had a

full complement of BH domains. Overall, these analyses suggest that the BCL-2 proteins of nematodes are different from those of other phyla and usually cluster to the exclusion of the well-defined functional BCL-2 protein sub-groups recognised in vertebrate taxa.

**Unique features of CED-4/APAF-1 proteins in nematodes.** Based on the *C. elegans* prototype, nematode BCL-2 proteins inhibit apoptosis by sequestering the apoptosome scaffold protein, CED-4, on mitochondria. Our sequence and structural analyses, however, suggest that apoptosis regulation differs among taxa within and among clades. Hence, we conducted an extensive phylogenetic analysis of CED-4 orthologues representing 89 taxa, together with APAF-1/CED-4 proteins representing members from diverse phyla including sponges, cnidaria, arthropods, platyhelminths, rotifers, annelids and chordates (Fig. 4, Supplementary Table 6 and Supplementary Data 3). The phylogenetic tree showed that CED-4/APAF-1 proteins of

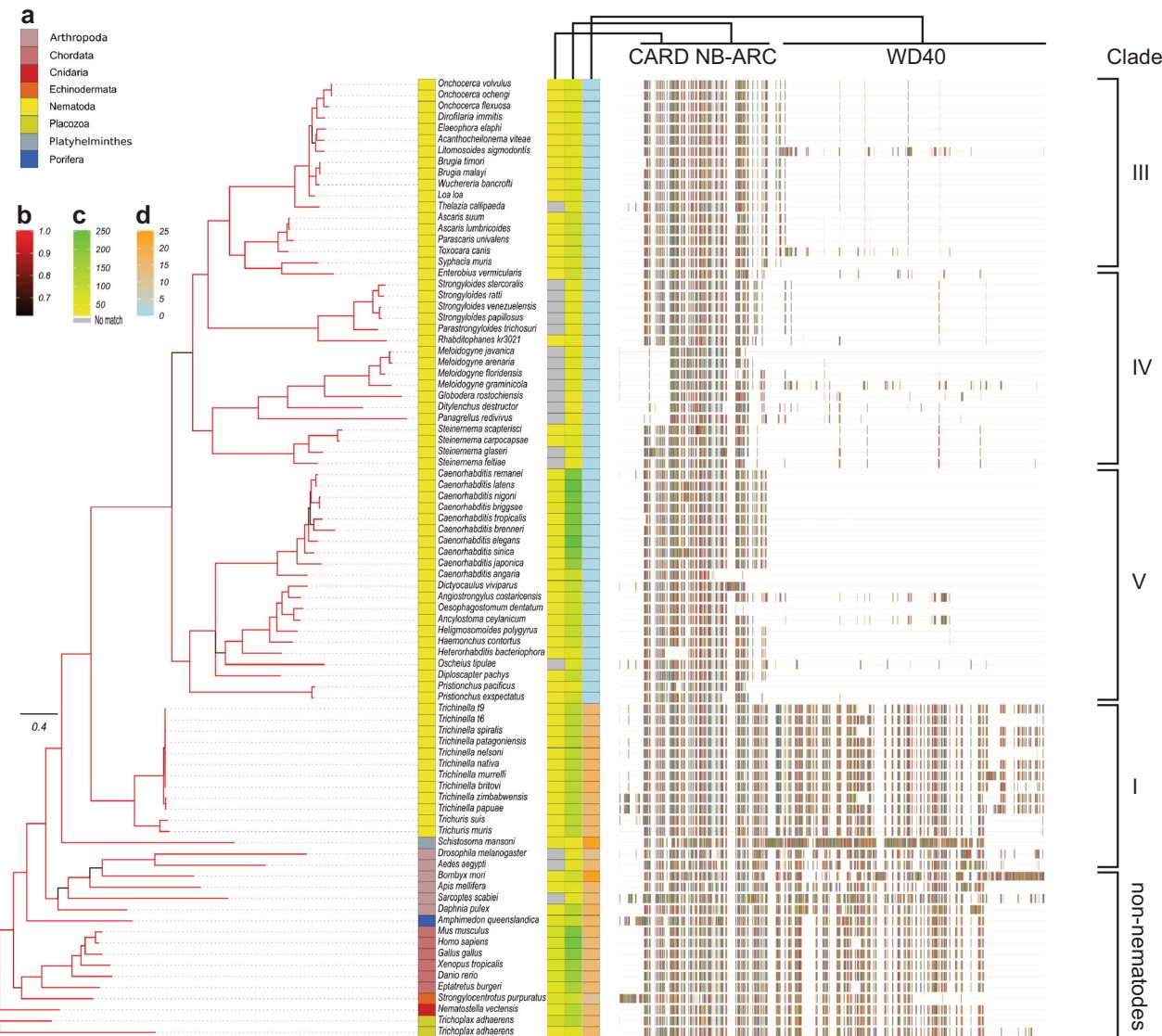

**Fig. 4 Relationships among predicted nematode CED-4/APAF-1 proteins and other eukaryotic taxa.** Consensus Bayesian phylogenetic tree of predicted nematode APAF-1 proteins and representative eukaryotic taxa. **a** Tree tips are labelled by their taxonomic grouping (phylum) and species name. **b** Branches are coloured by the posterior probability support values of adjacent nodes, where black is 0.5 and red is 1.0. **c** Bitscore of conserved sequence homology to CARD (Pfam: PF00619.21) and NB-ARC (Pfam: PF00931.22) domains are presented in colour scale (yellow low; green high; no sequence homology grey). **d** Number of predicted WD40 repeats in the C terminus is shown using a colour scale (zero: blue to high: orange). Aligned amino acid residues used for tree construction are also shown and major domains are highlighted. Tree scale represents 0.4 substitutions per site.

nematodes are all encoded by single-copy genes, and its topology was quite consistent with speciation[32]. CED-4 proteins usually clustered according to nematode clades, with the exception of nematodes in clade IV (*Meloidogyne, Globodera* and *Ditylenchus* of plants; *Steinernema* of insects, *Strongyloides* of mammals and the free-living *Panagrellus*). The proteins from the plant-parasitic taxa are conspicuous because of the lack of an identifiable caspase recruitment domain (CARD; Fig. 4) which is critical for apoptosome assembly. This information, together with an apparent absence of BCL-2 proteins, suggests a very distinct mechanism of apoptosis regulation in these taxa.

Another striking finding was that clade I nematodes possess multiple WD40-repeat motifs at the C terminus of the CED-4/APAF-1 protein (Fig. 4), similar to that observed in all phyla studied other than the Nematoda. The numbers of these repeats (15–17, depending on species) are consistent with those in the human orthologue that forms 7- and 8-blade β-propellers in the

assembled apoptosome. These findings for CED-4/APAF-1 proteins suggest that the apoptotic machinery of clades I and IV representatives is distinct from that of clades III and V nematodes studied here.

**Functionality of BCL-2 proteins representing clade I.** As our genome-guided data indicated distinct differences in clade I BCL-2 proteins, we selected *T. suis* as a representative species for functional studies due to the presence of two BCL-2 proteins (i.e. *T. suis* A and *T. suis* B) that could indicate potential different functions (i.e. pro- and anti-apoptotic). Previously, we successfully used $Bax^{-/-}/Bak^{-/-}$ mouse embryonic fibroblasts (MEF) to recapitulate the BCL-2-regulated apoptotic pathway from schistosomes[12] as this enabled any apoptotic responses to be attributed entirely to the reconstituted protein rather than endogenous BAX/BAK. We therefore reconstituted $Bax^{-/-}/Bak^{-/-}$ MEF with

both HA- and/or FLAG-tagged *T. suis* BCL-2 family proteins A and B, individually and in combination. Our focus was on testing whether either of these proteins was pro-apoptotic (BAX/BAK-like) as these proteins are absent from *C. elegans*.

The *T. suis* protein A expressed at levels similar to that seen for the same tagged human BAX and BAK, while expression of *T. suis* B was notably weaker (Fig. 5a, Supplementary Fig. 1a). Both proteins localised primarily to membrane-containing cellular fractions similar to BAK and BCL-2, rather than BAX that is found in both soluble and membrane fractions (Fig. 5b, Supplementary Fig. 1b).

In mammalian cells, pro-survival proteins directly bind pro-apoptotic BAX/BAK-like proteins, which can be readily detected by co-immunoprecipitation (co-IP), though the outcome can be influenced by whether CHAPS or Triton X-100 is used to lyse the cells[33]. Accordingly, we performed co-IPs on lysates from cells expressing HA- and FLAG-tagged *T. suis* protein A and B proteins prepared in both detergents. In both cases, no interaction was detected, unlike in a control co-IP where an interaction between HA-BAK and endogenous MCL-1 was readily detected in Triton X-100-lysed cells (Fig. 5c and Supplementary Figs. 2, 3).

Pro-apoptotic BCL-2 proteins interact with pro-survival proteins via their BH3 domain. Hence, to further investigate whether the *T. suis* proteins A or B proteins interact, we purified recombinant proteins and measured their affinity for synthetic peptides corresponding either to their own BH3 domain, or that from the other protein, using microscale thermophoresis (MST), as previously[34]. No obvious interaction ($K_D > 50\,\mu M$) was observed between either protein and the others' BH3 domain, or its own BH3 sequence, suggesting that these proteins do not regulate each other as seen for mammalian BCL-2 proteins (Supplementary Fig. 4). A positive control interaction between BCL-XL and the BAK BH3 domain showed binding as expected ($K_D$ 178±34 nM).

Subsequently, we performed functional assays to gain further insight into whether either protein could be pro-apoptotic. Expression of either *T. suis* protein A or B had no impact on $Bax^{-/-}/Bak^{-/-}$ cell survival, and the cells were readily cultured without a noticeable decline in viability. To determine whether additional stimuli was required to activate apoptosis, the cells were treated with etoposide that induced both BAX- and BAK-dependent cell death, but had no effect on cells expressing either or both *T. suis* proteins (Fig. 6a and Supplementary Data 4). To eliminate the possibility that apoptosis was being blocked due to strong interaction of the *T. suis* proteins with endogenous pro-survival proteins, we performed cytochrome *c* release assays on permeabilised $Bax^{-/-}/Bak^{-/-}$ cells expressing the *T. suis* proteins. Here, cells were treated with BIM BH3 peptide that can potently neutralise all mammalian pro-survival proteins. This resulted in cytochrome *c* translocation from pellet to soluble fractions in cells expressing BAX or BAK, but not either *T. suis* protein (Fig. 6b, Supplementary Fig. 5a). Combined, these cellular and biochemical studies suggest that in *T. suis*, neither BCL-2 protein engages the other nor do they have the capacity to act similarly to BAX and BAK to execute apoptosis.

Finally, we examined the BCL-2 protein from *P. sambesii* shown to cluster with vertebrate BAK-like proteins, together with those from other non-vertebrate species (*Helobdella robusta* and *Hydra vulgaris*) within the same grouping. Each protein was expressed in $Bax^{-/-}/Bak^{-/-}$ MEF and detected in both pellet and soluble fractions (like e.g. BAX) (Fig. 6c and Supplementary Fig. 5b), though predominantly in the mitochondria-containing pellet fraction. In cytochrome *c* release assays, there was no strong evidence for pro-apoptotic activity (Fig. 6d and Supplementary Fig. 5c). These data suggest that BCL-2 protein function cannot necessarily be predicted from sequence alone.

## Discussion

Biochemical and functional studies of BCL-2 family proteins in *C. elegans* laid the foundation for our understanding of how apoptosis is regulated in humans, despite the subsequent discovery of significant diversity between the pathways[24]. Here, we exploited the availability of a large number of nematode genome sequences to explore the BCL-2-regulated apoptosis pathway of nematode taxa representing four evolutionary clades. These analyses revealed unexpected diversity in critical pathway components.

The first obvious difference was that, unlike in *C. elegans*, more than one BCL-2 protein was identified for some nematode species within clades I and III. The sequence identity of BCL-2 para-logues in the clade III nematodes was significantly higher than those from the clade I nematodes, consistent with clade I being the basal of the two. This expansion of BCL-2-like proteins has a number of possible implications. Probably, the simplest inter-pretation is that multiple paralogues have evolved in some nematode taxa to "fine tune" their particular apoptotic response (s) to different stimuli as they adapt to differing environmental conditions (within or outside of their host/s) to establish and survive in their host(s) and/or to develop and reproduce. In this context, multiple BCL-2 proteins would be expected to be linked to intricate and taxon-specific control of the apoptotic response throughout the life cycle of a particular nematode. This aspect of apoptosis regulation in parasitic nematodes has not been explored previously and warrants detailed investigation.

A second possibility is that additional BCL-2-like proteins in some taxa might be pro-apoptotic BAX/BAK-like molecules, though in *T. suis*, we found no evidence that either BCL-2-like protein induces apoptosis, either basally or in response to a strong death signal. This conclusion is supported by biochemical data showing that these *T. suis* proteins do not interact with one another as occurs when both pro-survival and BAX/BAK proteins exist. Interestingly, our phylogenetic analysis revealed that one BCL-2 protein from *R. culculovirax* and *P. sambesii* clustered (with strong support: posterior probability 0.95) with BAK-like proteins from vertebrates and invertebrate taxa (Fig. 3), although studies on the *P. sambesii* protein, again, found no evidence that it was pro-apoptotic. Beyond specific positions[35], the sequence signature(s) discerning BAK/BAX versus pro-survival proteins remains elusive. Hence, it is unclear at this stage why these proteins segregated from the other nematode proteins. Notably, two other BCL-2 proteins representing cnidarians and annelids also clustered with the BAK proteins but did not have pro-apoptotic activity (Fig. 6d), further reinforcing the challenges with predicting BCL-2 protein function from primary amino acid sequence. One caveat to all of these studies is that they were conducted using a mammalian tissue culture system. While this has been used successfully to dissect the function of BCL-2 family members from other organisms, such as schistosomes[12], it is possible that, in some cases, the native physiological cellular environment might be required to reveal the pro- or anti-apoptotic activity of the proteins.

Intriguingly, CED-4/APAF-1 proteins inferred for *Trichuris* and *Trichinella* clade I nematode taxa all possess WD40 repeats at their C terminus, unlike all other nematode taxa, but similar to that observed in other phyla. In mammals, the WD40 domains bind cytochrome *c* released from mitochondria by BAX/BAK-like proteins which is necessary for the assembly of an apoptosome[23]. As *C. elegans* does not possess a BAX/BAK-like protein, apop-tosome formation in this species does not rely on cytochrome *c* release following mitochondrial permeabilisation. Notably, the CED-4/APAF-1 homologue in *Drosophila melanogaster* contains WD40 repeats and a BAX/BAK-like protein (DEBCL) but does not require cytochrome *c* binding for apoptosome assembly[36,37]. Hence, the presence of WD40 repeats does not necessarily mean

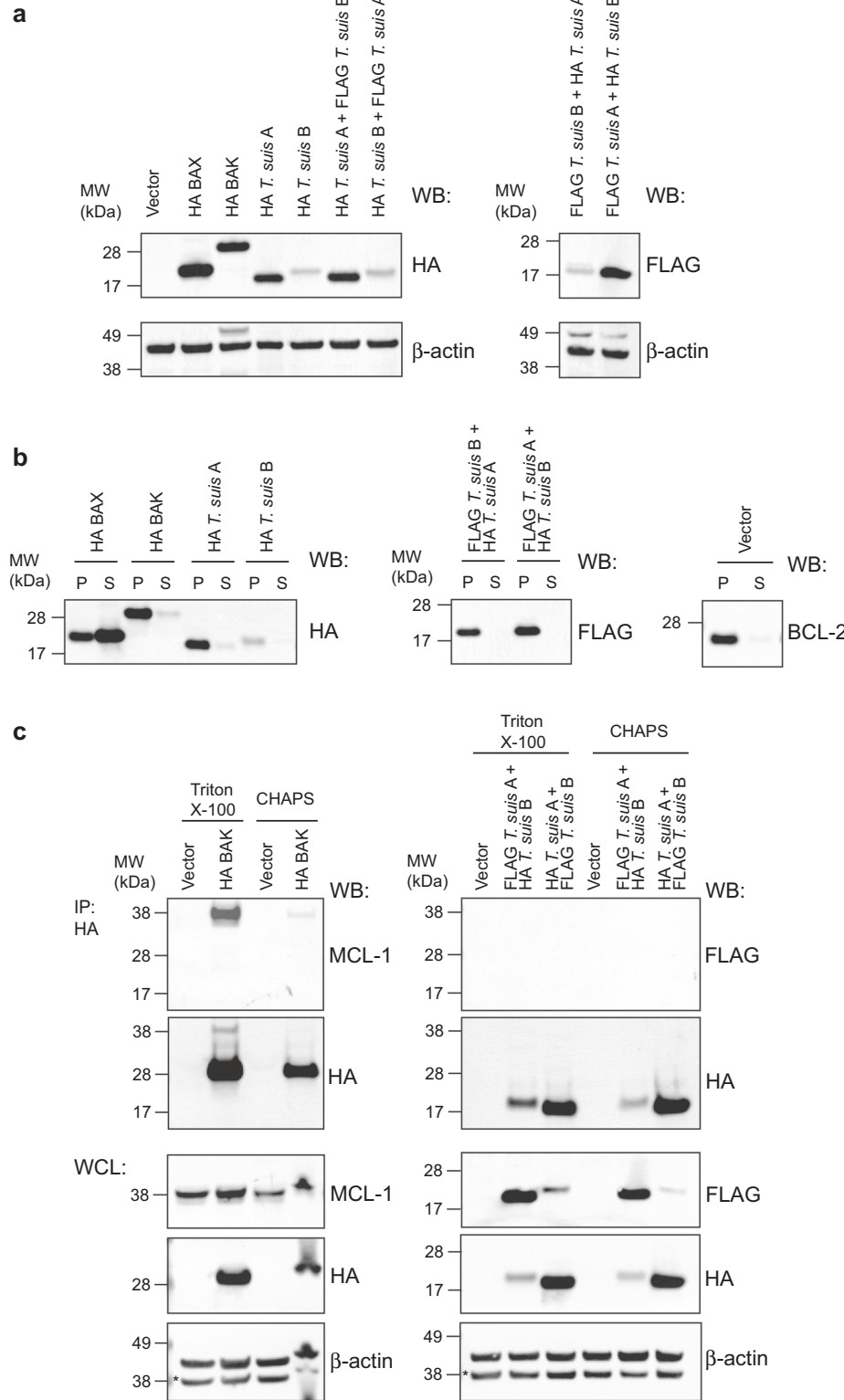

**Fig. 5 Expression, localisation and interactions of the *Trichuris suis* BCL-2-like proteins A and B. a** HA-tagged *T. suis* A expressed at similar levels to mammalian BAX and BAK when introduced into *Bax*$^{-/-}$/*Bak*$^{-/-}$ MEF, while expression levels of *T. suis* B was lower. **b** Both *T. suis* BCL-2-like proteins localised to the mitochondria-containing pellet fraction (P) similar to mammalian BAK and BCL-2, whereas BAX is localised in both cytosolic soluble (S) and pellet fractions. **c** HA- and FLAG-tagged *T. suis* A and B did not interact under any of the detergent lysis conditions tested unlike BAK and MCL-1 which co-immunoprecipitated in the presence of Triton X-100. WB: western blot, WCL: whole-cell lysate, IP: immunoprecipitation. Uncropped blots used to construct figures are provided in Supplementary Figs. 1–3.

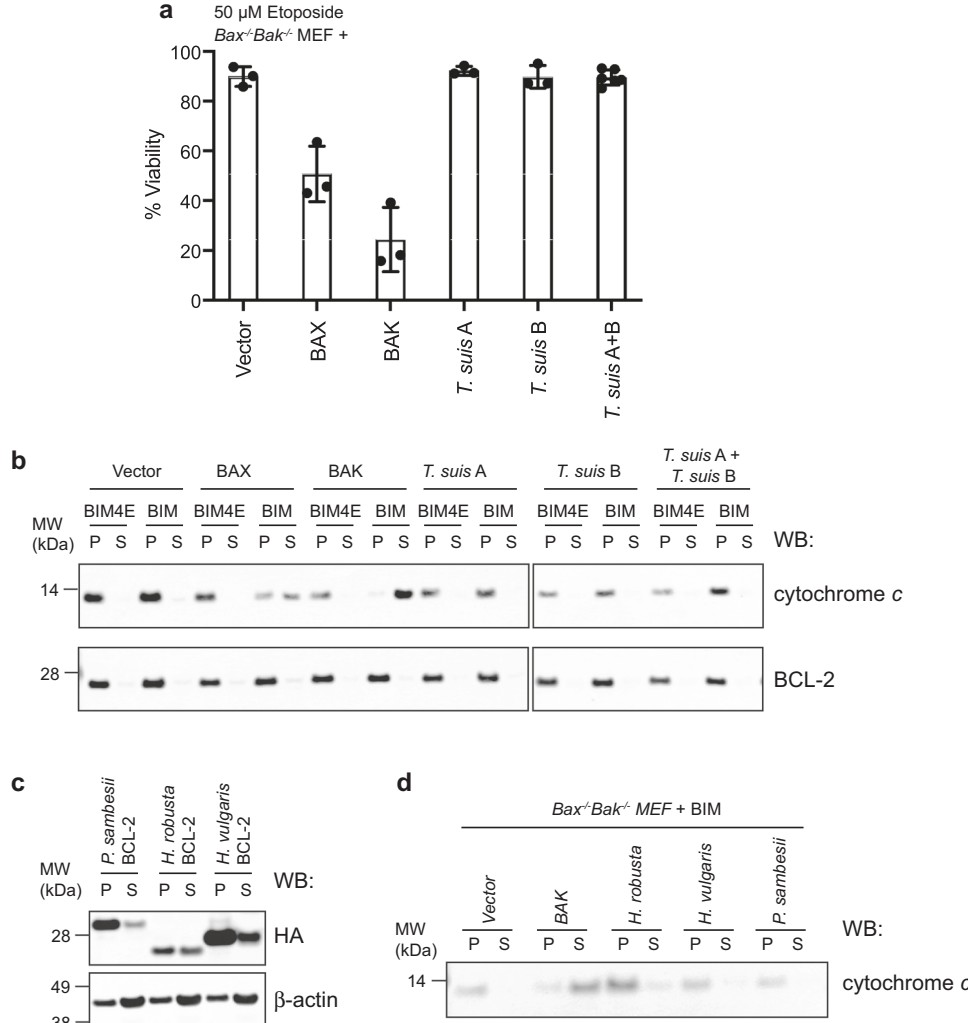

**Fig. 6 Both *Trichuris suis* proteins A and B do not exhibit pro-apoptotic activity unlike mammalian BAX or BAK.** The presence of *T. suis* A and/or B did not **a** restore the ability of cells deficient for BAX and BAK to undergo apoptosis following etoposide treatment unlike BAK or BAK (data are the mean ± standard deviation of $n = 3$ biologically independent experiments for all except *T. suis* A + B where $n = 6$ combining data from two separate cell lines expressing these proteins; raw data available in Supplementary Data 4) or **b** enable the release of cytochrome *c* from the mitochondria (P: mitochondria-containing pellet) into the cytosol (S: soluble fraction) following treatment with BIM BH3 peptide, unlike BAX and BAK. BIM4E is an "inactive" mutant BIM BH3, negative control peptide. Blots were re-probed with anti-BCL-2 (an exclusively membrane-bound protein) antibody as a fractionation control. **c** Nematode (*Plectus sambesii*) and non-vertebrate (*Helobdella robusta* and *Hydra vulgaris*) BCL-2 proteins that clustered with BAK proteins in the phylogenetic analysis (Fig. 4) were expressed in $Bax^{-/-}/Bak^{-/-}$ MEF and localised in both the mitochondria-containing pellet (P) and cytosolic soluble (S) fractions, similar to mammalian BAX. **d** These BCL-2 orthologues do not have significant pro-apoptotic capacity as demonstrated by their inability to release cytochrome *c* from the mitochondria (P) fraction into the cytosol (S) fraction following treatment with BIM BH3. Uncropped blots used to construct **b**–**d** are provided in Supplementary Fig. 5.

that cytochrome *c*, and by analogy to the mammalian pathway, a BAX/BAK protein, is required for apoptosome formation, consistent with our observations that neither of the two *T. suis* BCL-2-like proteins could induce cytochrome *c* release when reconstituted into mammalian cells. Curiously, the loop following helix α4 in CED-9 which is required for release from CED-4 is absent from BCL-2 proteins representing clade I nematodes, as is the case in mammalian BCL-2 proteins which do not directly engage APAF-1. While this observation does not discount the possibility that BCL-2 proteins of clade I nematodes inhibit apoptosome formation in the same manner as occurs in *C. elegans*, it does suggest that the molecular details of the "release" mechanism are distinct from the free-living nematode.

The absence of BCL-2-like proteins from nematodes which are endoparasites of plants was also intriguing. This observation suggests that their gene sequences have diverged to an extent that

the characteristic BCL-2 "signature" is no longer apparent, as reported for several viral BCL-2 proteins (e.g., M11L)[38]. In this context, BCL-2 proteins of nematode taxa within the same clade (e.g., clade IV) were more diverse compared with those representing other clades. These plant or insect nematodes also have an unusual CED-4-like protein that lacks a recognisable CARD; hence, plant-parasitic nematodes likely regulate apoptosis in a unique way as compared with other nematode taxa investigated here and, indeed, any other phyla studied to date.

While, from the outset, our aim was not to reconstruct the phylogeny of nematode species, the topologies of the trees constructed using sequence data were relatively consistent with the phylogenetic relationships of the species. For the analyses, only single-copy orthologous genes were used to avoid inconsistencies caused by the presence of one or more paralogue[39]. Hence, the presence of paralogous BCL-2 proteins, not unexpectedly, led to

inconsistencies with established nematode species trees[32] compared with the tree constructed using APAF-1-like protein sequences inferred from single-copy genes from individual nematode species. Other factors contributing to inconsistencies include the absence of BCL-2 and/or APAF-1-like protein genes from some taxa and the presence of rapidly evolving sites in such genes, for example, in species of *Strongyloides*[40].

In conclusion, our findings reject the hypothesis that the *C. elegans* prototype for the regulation and execution of the intrinsic apoptosis pathway represents all members of the phylum Nematoda. The important structural differences between both CED-9/BCL-2-like and CED-4/APAF-1-like proteins in species of basal nematode lineages suggest that key alterations occurred in the radiation of the Secernentea (sensu Chittwood[27,28]). There are also notable differences among nematode taxa within both adenophorean and secernentean groups, including BCL-2 gene expansions in some clade I (Adenophorea) and III (Secernentea) taxa, but not in others, and a loss of recognisable BCL-2-like proteins from some plant nematodes. Nevertheless, given the deep phyletic separation of nematodes from other groups—estimated to be 1100 to 1200 million years ago[41]—and the unparalleled diversity in morphology, biology, behaviour and environmental niches in which they are found, it is perhaps not surprising that nematode cells have adopted different mechanisms to facilitate their individual survival responses to diverse cellular stresses and developmental cues.

## Materials and methods

**Protein data sets**. All proteomes of nematode taxa and taxa representing extant lineages were obtained from Wormbase:Parasite (Release 12)[42] and ENSEMBL Metazoa (release 42)[43] databases, respectively (see Supplementary Table 1).

**Identification of BCL-2-like proteins**. For each data set, proteins with sequence homology to the BCL-2 family (Pfam: PF00452.19)[44] were identified using *hmmsearch* (HMMER v.3.2.1; http://hmmer.janelia.org/)[45] with an E-value threshold of 1E$^{-04}$. Proteins containing a conserved BCL-2 domain (spanning from the BH4 to the BH2 domain) were aligned (trimmed alignment length of 204 amino acids) and subjected to manual curation of individual BH domains with reference to respective recognised consensus sequences. Identical proteins were removed using CD-HIT (v.4.7)[46] with option -c 1.0. The predicted BCL-2 proteins of nematodes and select taxa were aligned (trimmed alignment length of 185 amino acids) using the PRO-MALS3D software[47] with resolved BCL-2 tertiary structures for *C. elegans* (CED-9; PDB ID# 1OHU), *T. suis* (PDB ID# 6V4M), *Mus musculus* MCL-1 (PDB ID# 1WSX Chain A)[48], *M. musculus* BCL-XL (PDB ID# 1PQ0 Chain A)[49], *Homo sapiens* BCL-2 (PDB ID# 1G5M Chain A)[50], *H. sapiens* BAX (PDB ID# 1F16, Chain A)[51], *H. sapiens* BFL-1 (PDB ID# 5WHI Chain A)[52], *H. sapiens* BAK (PDB ID# 2YV6 Chain A)[53], *Gallus gallus* BOK (PDB ID# 5WDD Chain A)[54]. The alignments were improved using the programme MUSCLE v.3.7 (-refine option)[55] and trimmed using trimAl v.1.4.1 (-gappyout option)[56]. Sequences were aligned and then subjected to Bayesian inference (BI) analysis using the programme MrBayes (v.3.2.6)[57]. Posterior probabilities (pp) were calculated using an LG model with fixed rate matrices, generating 2,000,000 (nematode only) or 16,000,000 (representative eukaryote) trees and sampling every 200th tree until potential scale reduction factors for each parameter approached one. The initial 25% (nematode only) or 50% (representative eukaryote) of trees were discarded as burn-in, and the others were used to construct a majority rule tree. Phylogenetic trees were rendered and annotated using ggtree (v.1.10.5)[58] in R (v.3.4.3; http://www.R-project.org/).

**Identification of APAF-like proteins**. For each data set, APAF-1-like proteins with domains homologous to CARD (Pfam: PF00619.21), NB-ARC (Pfam: PF00931.22) and WD40 (Pfam: PF00400.32) Pfam domains[44] were identified using *hmmsearch* (HMMER v.3.1b1; http://hmmer.janelia.org/) with a default E-value threshold of 1E$^{-05}$, and curated manually. To increase the sensitivity of identification of APAF, CARD and NB-ARC domains, we constructed a specific HMM for APAF-like proteins from an alignment of identified proteins with a conserved CARD, NB-ARC and at least one WD40 Pfam domain. These proteins were aligned using the programme MAFFT (v7.215)[59] employing the L-INS-i option, and an HMM was constructed using aligned sequences within the CARD and NB-ARC domains using *hmmbuild* (HMMER v.3.2.1; http://hmmer.janelia.org/). Subsequently, we identified APAF-like proteins with sequence homology to the constructed APAF domain using *hmmsearch* (HMMER v.3.2.1; http://hmmer.janelia.org/) with a default E-value threshold of 1E$^{-10}$, combined with manual curation. For each taxon, the protein sequence with greatest amino acid sequence

homology to the APAF HMM domain was inferred to be an APAF-like protein. Proteins without a Pfam NB-ARC domain, with a high proportion of ambiguous sequence regions (denoted as Xs in the conceptually translated protein), predicted to be a partial gene model or identified as identical using CD-HIT (v.4.7)[46] were removed using the option -c 1.0. The representative proteins were then aligned (alignment length of 985 amino acids) using the programme MAFFT (v7.215)[59] employing the L-INS-i option and specifying a BLOSUM55 substitution matrix using the –aamatrix option. Aligned sequences (excluding the WD40 domains) were then subjected to BI analysis using the programme MrBayes (v.3.2.6) as described above for identification of the BCL-2-like proteins. Within each APAF-like protein, WD40 repeats were identified using WDRR (v.1.0)[60], using predicted secondary structure information and conserved template to profile alignment algorithm with a combined scoring function of BLOSUM62 and dot product[60]. Phylogenetic trees were rendered and annotated as described above using 4,000,000 generations and discarding the initial 25% of trees as burn-in as for the BCL-2 proteins.

**Cell culture**. MEF were cultured in Dulbecco's modified Eagle's Medium high-glucose (Thermo Fisher Scientific) media supplemented with 10% (v/v) foetal calf serum (PAA Laboratories), 250 μM L-asparagine (Sigma-Aldrich), 50 μM 2-mercaptoethanol (Sigma-Aldrich), 1 mM HEPES (Thermo Fisher Scientific), 100 U/mL penicillin and 100 mg/mL streptomycin (Gibco) and incubated at 37 °C with 10% $CO_2$.

**Generation of cell lines**. Phoenix Ecotropic packaging cells were transiently transfected with either pMiG or pMiCh (*MSCV-I*RES-GFP or m*Che*rry) retroviral constructs expressing HA- or FLAG-tagged genes of interest, using X-tremeGENE DNA transfection reagent (Sigma-Aldrich). Following incubation for 24 h at 37 °C with 10% $CO_2$, culture medium was replaced with MEF culture medium and incubated at 32 °C with 10% $CO_2$ for a further 24 h. Virus-containing supernatant was then harvested the following day and filtered through a 0.45-μM filter (Millipore) to remove cell debris. One day prior to retroviral infection, $Bak^{-/-}/Bak^{-/-}$ MEF (obtained from the Walter and Eliza Hall Institute and confirmed to be mycoplasma negative) were plated into six-well plates and then spin-infected with filtered virus-containing supernatants by centrifugation at 2500 r.p.m. for 45 min at 32 °C in the presence of 4 μg/mL polybrene (Sigma-Aldrich). The medium was replaced with MEF culture media 24 h following the spin-infection. Following growth for at least 3 days, transduced cells were selected for mCherry- or GFP-positivity using fluorescence-activated cell sorting.

**Fluorescence-activated cell sorting-based apoptosis assay**. Cells were seeded into 24-well plates and, 24 h later, were treated with 50 μM Etoposide (Sigma-Aldrich) for 24 h. Both floating and adherent cells were then collected and washed with binding buffer (BD Bioscience) before staining them with Annexin V-APC (BD Biosciences) and propidium iodide (Sigma-Aldrich), and subsequently analysed using a flow cytometer (BD FACSCanto II, BD Biosciences). Cell viability data were analysed using FlowJo, with the percentage of viable cells following treatment (Annexin V$^{-ve}$/propidium iodide$^{-ve}$) calculated relative to the percentage of viable cells cultured in vehicle only.

**Preparation of cell lysates**. $Bak^{-/-}/Bak^{-/-}$ MEF cells expressing proteins were pelleted and incubated for 1 h at 4 °C in lysis buffer (20 mM Tris-pH 7.4, 135 mM NaCl, 1.5 mM $MgCl_2$, 1 mM EGTA, 10% (v/v) glycerol containing either 1% (v/v) Triton X-100 (Sigma-Aldrich) or 1% (w/v) CHAPS (Sigma-Aldrich), supplemented with a cocktail of protease inhibitors (Complete Mini, EDTA-free; Roche). Following cell lysis, insoluble material was removed by centrifugation prior to use.

**Co-IP experiments**. Cell lysates were incubated with either 50 μL anti-HA-conjugated (Clone 3F10, Roche) or anti-FLAG-conjugated (Clone M2, Millipore) agarose resin for 1 h at 4 °C, with mixing. The beads containing the captured antibody–antigen complex were pelleted and then washed four times with lysis buffer. After the final wash, the bound proteins were eluted by boiling the beads in the presence SDS-PAGE sample-loading buffer.

**Cytochrome c release assay**. $Bak^{-/-}/Bak^{-/-}$ MEF cells expressing proteins were pelleted and permeabilised by incubation in permeabilisation buffer (0.05% (w/v) digitonin (Calbiochem) in 20 mM HEPES pH 7.2, 100 mM KCl, 5 mM $MgCl_2$, 1 mM EDTA, 1 mM EGTA, 250 mM sucrose, supplemented with protease inhibitors; Roche) for 5 min on ice. Permeabilisation was monitored by trypan blue staining. Once >90% of cells stained positively for trypan blue, the mitochondria-containing crude lysates were incubated with 10 μM peptides corresponding to the BH3 domain of BIM (DMRPEIWIAQELRRIGDEFNAYYARR) or an inactive BIM mutant, BIM4E (DMRPEIWEAQEERREGDEENAYYARR) for 1 h at 30 °C prior to pelleting. The supernatant was retained as the soluble fraction, while the pellet, which contained unpermeabilised mitochondria, was solubilised in the lysis buffer containing 1% (v/v) Triton X-100, as described above.

**Western blot analysis**. Total protein extracts or eluted co-immunoprecipitated proteins were resolved by SDS-PAGE and transferred onto nitrocellulose membranes. In brief, membranes were incubated in 5% (w/v) skimmed-milk powder in phosphate-buffered saline, pH 7.4 for a minimum of 1 h at room temperature. Blocked filters were then probed with antibodies to HA (Clone 12CA5, Roche, 1:1000 dilution), FLAG (Clone M2, Sigma-Aldrich, 1:2000 dilution), BCL-2 (Clone 7/BCL-2, BD Biosciences, 1:500 dilution), β-actin (Clone AC-74, Sigma-Aldrich, 1:5000 dilution), cytochrome *c* (Clone 7H8.2C12, BD Pharmingen, 1:1000 dilution).

**Expression and purification of recombinant proteins**. For crystallography and binding studies, recombinant *T. suis* protein A (ID# BCL_199) and protein B (ID# BCL_257) (Supplementary Table 2) were produced from synthetic genes codon-optimised for expression in *Escherichia coli* (from Genscript, USA) and with an N-terminal 6xHistidine tag and C-terminal truncations (i.e. *T. suis* A ΔC28 and *T. suis* B ΔC26). Recombinant proteins were produced in *E. coli* following induction of expression for 3 h at 37 °C and then purified as described previously for schistosome BCL-2 protein[12]. Hexa-histidine tagged recombinant BCL-XLΔC24 for MST experiments was expressed and purified as described previously[61].

**Microscale thermophoresis**. Affinity measurements by MST were performed using a Monolith NT.115 instrument (NanoTemper Technologies), as described previously[34], employing purified, recombinant *T. suis* A ΔC28, *T. suis* B ΔC26 and BCL-XLΔC24 labelled using the NHS RED NanoTemper labelling kit (cat. no. MO-L001) according to the manufacturer's instructions. For this assay, each labelled protein was mixed with unlabelled synthetic peptide (either *T. suis* protein A (BH3: SVRAPIRLCDAMFLMCDLFERKFHDR) or *T. suis* protein B (BH3: PILPEVNKTALCMRAMGEVFEDRYKT) or for the positive control the interaction between BCL-XL and BAK, BAK (BH3: PSSTMGQVGRQLAIIGDDINRRY DSE)). All peptides were synthesised to >90% purity (Mimotopes, Australia) and diluted in phosphate-buffered saline. All experiments were incubated for 30 min before applying samples to Monolith NT standard-treated capillaries (Nano-Temper Technologies). Thermophoresis was measured at 25 °C with laser off/on/off times of 5 s/30 s/5 s. Experiments were conducted at 40% LED power and 40% MST infrared laser power. Data from three independently performed experiments were fitted to the single binding model using NT.Analysis software 1.5.41 (NanoTemper Technologies) the signal from thermophoresis + T-Jump.

**Protein crystallisation**. Crystallisation experiments (assessing approximately 1000 conditions) were performed at the Bio21 Collaborative Crystallisation Centre. Crystals of *T. suis* protein A ΔC28 were grown using the sitting drop method at room temperature in 0.1 M Tris-chloride (pH 8.5), 0.2 M lithium sulfate and 2 M sodium chloride. Prior to cryo-cooling in liquid nitrogen, crystals were equilibrated into a cryoprotectant consisting of reservoir solution containing 15% (v/v) ethylene glycol. Crystals were mounted directly from the drop and plunge-cooled in liquid nitrogen.

**Collection of diffraction data and structure determination**. Diffraction data were collected at 100 K at the Australian Synchrotron MX2 beamline (Victoria, Australia). The diffraction data were integrated with XDS[62]. A search model (PDB: 3FDL Chain A from the structure of BCL-XL in complex with BIM[63]) for molecular replacement was first identified using AUTORICKSHAW[64,65], and then the initial model was obtained using PHASER[66,67]. Multiple rounds of building in COOT[68] and refinement in PHENIX[62] led to the final models.

**Reporting summary**. Further information on research design is available in the Nature Research Reporting Summary linked to this article.

## Data availability
All sequence data used for phylogenetic analyses are available from publicly available databases (Wormbase: Parasite (Release 12) 40 and ENSEMBL Metazoa (release 42)). Details of accession numbers of all sequences used for analysis are listed in the Supplementary Tables 1, 2 and 6 and provided as Excel spreadsheets in Supplementary Data 1, 2 and 3. All data used to determine the crystal structure have been deposited in the PDB (Accession number 6V4M). Raw data for cell killing assay (Fig. 6a) are provided in Supplementary Data 4. Any other data are available on request from the authors.

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

## Acknowledgements

Research funding from the Australian Research Council (LP180101334 to N.D.Y., LP180101085 to R.B.G., DE190100806 to T.P.S.d.C. and FT150100212 to E.F.L) is gratefully acknowledged. T.P.S.C. also acknowledges support from the National Health and Medical Research Council of Australia (GNT1091976). We also thank the La Trobe University Comprehensive Proteomics Platform for providing infrastructure support.

## Author contributions

N.D.Y., E.F.L and W.D.F. designed the study. T.J.H., M.E. and S.T. performed the biological experiments. T.P.S.d.C. performed the MST experiments. N.J.K., B.J.S., E.F.L. and W.D.F. collected X-ray diffraction data and/or solved and analysed the crystal structure. N.D.Y., M.W., E.F.L. and W.D.F. identified sequences. N.D.Y. and M.W. performed phylogenetic analyses. N.D.Y., R.B.G., E.F.L. and W.D.F. analysed all the data and wrote the manuscript.

## Competing interests

The authors declare no competing interests.
