## [Peer Review File · Communications Biology]

Reviewers' comments:

Reviewer #1 (Remarks to the Author):

I have been asked to explicitly review components of the manuscript that utilized genomic analyses. Thus, I have restricted my comments and suggestions to only those sections.

In Diversity in the Intrinsic Apoptosis Pathway of Nematodes, Neil D. Young and colleagues examined the diversity of the apoptosis pathway across 89 nematodes. They found variation in the copy number of BCL-2 sequences across species. Using Bayesian molecular phylogenetic techniques, they inferred the evolutionary history of BCL-2 genes. A similar approach was used for putative CED-4/APAF-1 orthologs. The authors then examined domain presence and absence coupled to functional studies to further examine diversity of BCL-2 sequences.

Overall, I liked the manuscript and appreciate the authors ability to conduct both wet- and dry-lab experiments. That being said, I do have some concerns which largely revolve around the lack of details in the methods section.

Main comments

1) In the methods, many details necessary for reproducibility are missing. Here is a list of questions I have that can easily be clarified by expanding on the methods presented in the manuscript.

i) What sequence similarity thresholds were used when using hmmsearch?

ii) What parameters were used when removing identical proteins with CD-HIT? Furthermore, what is the justification of doing so? If a gene has been recently duplicated, it is reasonable to think that there could be two identical copies.

iii) Please clarify what BLOSUM substitution matrix was used during sequence alignment with Mafft. There is no BLOSUM55 matrix in the Mafft documentation (or any BLOSUM55 that I am aware of). I believe this is a simple typo.

iv) When conducting phylogenetic analysis, alignment length can heavily influence inferences. What was the length of the alignments used as input for phylogenetic inference?

2) Regarding the phylogenetic analyses, I think the significance of the various topologies can be further emphasized by discussing concordance (or lack thereof) between the species tree and the gene tree.

3) A lot of the analysis relies on the comparison of the various evolutionary clades. However, some of the bipartitions that differentiate the clades (such as III, IV, and V in Figure 2) have poorly supported backbones. I think it would be worth it to describe how well or poorly supported these bipartitions are and whether or not there is another well supported topology among the sampled phylogenies.

4) In the discussion section, there is a paragraph dedicated to the absence of genes, specifically BCL-2. I think the manuscript would benefit from discussion about duplication of genes and what they may mean for the evolvability of the apoptosis pathway.

Minor comments

1) The PFam domain for the BCL-2 domain does not match the other text. Please edit to provide the full PFam domain accession ID.

2) Why were the WD40 domains not used for BI analysis for APAF-like proteins?

Reviewer #2 (Remarks to the Author):

Young et al. investigated the presence of BCL-2-like genes in free-living and parasitic nematode taxa. The BCL-2 is involved with the regulation of the apoptosis process. The authors found different BCL-2 orthologues in the investigated taxa and identified that the basal clade I CED-4/APAF-1 lacks a functionally important feature of the *C. elegans* orthologue and also possess WD40-repeat sequences associated with apoptosome assembly, not present in *C. elegans*. The manuscript is well-written, show relevant results, and the conclusion is original. I suggest minor revision before publishing.

In the Material and Methods session ("Identification of APAF-Like Proteins" session), different tools are cited by their electronic addresses, but it is highly recommended to cite the original references for each tool. See HMMER v.3.2.1 and MrBayes (v.3.2.6).

Page 5, 3rd paragraph. Please correct the scientific name of *Globodera*

Please be careful with the reference format. In different parts of the text, the references citations are not according to the journal requirements.

In supplementary material, the authors informed incorrectly the name of the journal.

The authors state that that the two BCL-2 proteins are similarly based on the structural alignment, but it is recommended to uses a structural similarity metric, such as TM-score or RMSD-Ca. In Figure 1 please cite the RMSD value and identify the N- and C- terminal region of both compared structures (panel B).

Reviewer #3 (Remarks to the Author):

Young et al. present an interesting report analyzing the apoptotic regulators in a broad range of nematode species. They describe the BCL-2 orthologues from free-living and parasitic nematode taxa and found a number of species possessing multiple BCL-2 like sequences, unlike the well-studied *C. elegans*, which contains one. By focusing the strain *T. suis*, they first solved the crystal structure of one of its BCL-2 orthologues and found that it lacks a motif present in CED-9 that is involved in binding CED-4, and then found no evidence that the two BCL-2 like proteins from *T. suis* interact with each other, suggesting that neither of the two proteins act as a BAX/BAK protein. Overall, this report indicates that the mechanism of apoptotic regulation in other nematodes (like *T. suis*) can differ significantly from the well-studied *C. elegans* strain.

The story is well written and interesting. I do have a some suggestions regarding the manuscript.

1. Figures 5 and 6 are well presented and contain proper positive controls necessary due to convincingly demonstrate the negative results regarding lack of physical interactions between *T. suis* A and *T. suis* B, and the functional apoptotic assays. However, I do believe that the conclusion present

on page 7 should present the limitation that these studies were done within a mammalian cell culture system. It is still possible that in its proper in vivo context, T. suis A and T. suis B may interact and/or act like BAX/BAK to promote apoptosis.

2. At the bottom of page 6, the authors describe experiments using synthetic peptides and recombinant proteins to further investigate the potential interaction between T. suis A and B. This data is not presented and should be (perhaps as a supplemental) if the authors deem it worthwhile to include in the text.

3. The authors when describing the crystal structures at the bottom of page 3 and top of page 4 could make it clearer in the text from that they solve the structure of one of the two T. suis BCL-2 orthologues and that the one they solve is referred to as T. suis A. The second T. suis form is not mentioned until later and the fact that the structure is T. suis A is apparent Figure 2 legend, but I believe that this should be made clearer earlier in the text.

REVIEWER 1

1 i) *What sequence similarity thresholds were used when using hmmsearch?*

E value thresholds for hmmsearch have been included in the materials and methods section for both the BCL-2 and APAF-1 alignments.

Response:

Changes in ms:

p.9 "...were identified using *hmmsearch* (HMMER v.3.2.1; <http://hmmer.janelia.org/>) with an E-value threshold of $1E^{-04}$."

p.10 "...with a default E-value threshold of $1E^{-05}$, and curated manually."

p.10 "...with a default E-value threshold of $1E^{-10}$, combined with manual curation."

1 ii) *What parameters were used when removing identical proteins with CD-HIT?*

Furthermore, what is the justification of doing so? If a gene has been recently duplicated, it is reasonable to think that there could be two identical copies.

Response:

Identity thresholds used for CD-HIT have now been included in the Materials and Methods section (see below). They were all set at 100% identity. Exact copies were removed as: 1) In draft genomes (common in the nematode genomics databases), problems of haplotype phasing and/or misassembly can result in exact copies of a gene that is incorrectly identified as a gene duplication event. As we report, there was significant nucleotide and encoded amino acid differences between the duplicated genes; and 2) It is common to remove exact copies of a sequence prior to running phylogenetic analyses so that false groups are not inferred and displayed.

Changes in ms:

p.10 "...removed using CD-HIT (v.4.7) ⁴³ with option -c 1.0."

p.10 "...were removed using the option -c 1.0"

1-iii) *Please clarify what BLOSUM substitution matrix was used during sequence alignment with Mafft. There is no BLOSUM55 matrix in the Mafft documentation (or any BLOSUM55 that I am aware of). I believe this is a simple typo.*

Response:

MAFFT permits the use of alternative substitution matrices using the -amatrix option. We checked several BLOSUM matrices and found BLOSUM55 to produce the most robust sequence alignment.

Change in ms:

p.10 "...using the program MAFFT (v7.215) ⁵⁶ employing the L-INS-i option and specifying a BLOSUM55 substitution matrix using the -amatrix option."

1-iv) *When conducting phylogenetic analysis, alignment length can heavily influence inferences. What was the length of the alignments used as input for phylogenetic inference?*

Response:

Length of the aligned sequences used for phylogenetic analyses have been added to the results section.

Changes in ms:

p.10 “were aligned (trimmed alignment length of 204 amino acid sequences)”

p.10 “...were then aligned (trimmed alignment length of 185 amino acid sequences)...”

p10 “representative proteins were then aligned (alignment length of 985 amino acid sequences) using the...”

2) Regarding the phylogenetic analyses, I think the significance of the various topologies can be further emphasized by discussing concordance (or lack thereof) between the species tree and the gene tree.

Response:

Thank you for this suggestion. We have added the following paragraph to the Discussion.

Change in ms:

p.9:

Whilst, from the outset, our aim was not to reconstruct the phylogeny of nematode species, the topologies of the trees constructed using sequence data were relatively consistent with the phylogenetic relationships of the species. For the analyses, only single copy orthologous genes were used to avoid inconsistencies caused by the presence of one or more paralogues³⁹. Hence, the presence of paralogous BCL-2 proteins, not unexpectedly, led to inconsistencies with established nematode species trees³² compared with the tree constructed using APAF-1-like protein sequences inferred from single-copy genes from individual nematode species. Other factors contributing to inconsistencies include the absence of BCL-2 and/or APAF-1-like protein genes from some taxa and the presence of rapidly-evolving sites in such genes, for example, in species of *Strongyloides*⁴⁰.

References added

39. Burki F, Roger AJ, Brown MW, Simpson AGB. The New Tree of Eukaryotes. Trends Ecol Evol. 2020;35(1):43-55.

40. Delsuc F, Brinkmann H, Philippe H. Phylogenomics and the reconstruction of the tree of life. Nat Rev Genet. 2005;6(5):361-75.

3) A lot of the analysis relies on the comparison of the various evolutionary clades. However, some of the bipartitions that differentiate the clades (such as III, IV, and V in Figure 2) have poorly supported backbones. I think it would be worth it to describe how well or poorly supported these bipartitions are and whether or not there is another well supported topology among the sampled phylogenies.

Response:

This was addressed by the additional paragraph added to the Discussion section (see response to point 2 above). With regards the topology, the MrBayes phylogenetic trees reported were the converged trees derived from 2 million, 4 million or 16 million generations depending on the tree (see methods and changes to ms below). Convergent diagnostics reported that

Potential Scale Reduction Factors (PSRF) for each parameter had approached one and the final tree was the best supported topology derived from our alignment. The nodal support for branches related to gene duplication and species divergence were robust. As discussed above, deriving a robust species tree with well supported taxonomic clade assignments was not the aim of this study and has now been discussed.

Changes to ms:

p.10: "...2,000,000 (nematode only) or 16,000,000 (representative eukaryote) trees and sampling every 200th tree until potential scale reduction factors for each parameter approached one. The initial 25% (nematode only) or 50% (representative eukaryote)...

p.11: "...using 4,000,000 generations and discarding the initial 25% of trees as burn-in as for the BCL-2 proteins...."

4) In the discussion section, there is a paragraph dedicated to the absence of genes, specifically BCL-2. I think the manuscript would benefit from discussion about duplication of genes and what they may mean for the evolvability of the apoptosis pathway.

Response:

We did in fact discuss this point at length – please refer to paragraphs 2 and 3 of the Discussion section.

Minor comments

1) The PFam domain for the BCL-2 domain does not match the other text. Please edit to provide the full PFam domain accession ID.

Thank you for identifying this. We now use consistent accession number formats for all Pfam HMM accession numbers.

Change to ms:

p.9 : "...homology to the BCL-2 family (Pfam: PF00452.19)"

2) Why were the WD40 domains not used for BI analysis for APAF-like proteins?

Response:

As the WD40 domain was not encoded in the majority of the aligned nematode APAF-like sequences, we elected to exclude it for phylogenetic analysis. We do not consider its absence to have had a significant impact on tree topology.

REVIEWER 2

1) In the Material and Methods session ("Identification of APAF-Like Proteins" session), different tools are cited by their electronic addresses, but it is highly recommended to cite the original references for each tool. See HMMER v.3.2.1 and MrBayes (v.3.2.6).

Response:

Mr Bayes was cited in the text (when first mentioned). We now include a reference for HMMER at first mention (p.9 - now reference 45).

Change to ms:

p.9 ref 45 now cited at first mention of HMMER

p15 Eddy, S. R. Accelerated profile HMM searches. *PLoS Comput Biol* **7**, e1002195, (2011) added to reference list

2) *Page 5, 3rd paragraph. Please correct the scientific name of Globodera*

Response:

Thank you for identifying this error. This has been corrected.

3) *Please be careful with the reference format. In different parts of the text, the references citations are not according to the journal requirements.*

Response:

All references have been checked and we believe them in accordance with the journal requirements.

4) *In supplementary material, the authors informed incorrectly the name of the journal.*

Response:

Change to ms:

p.13: This has now been corrected.

5) *The authors state that that the two BCL-2 proteins are similarly based on the structural alignment, but it is recommended to uses a structural similarity metric, such as TM-score or RMSD-C α . In Figure 1 please cite the RMSD value and identify the N- and C- terminal region of both compared structures (panel B).*

Response:

The structural similarity versus canonical BCL-2 family protein structure (BCL-XL) has been included in the legend to figure 1. The N-terminus is obscured (at the back) of all those figures so could not be labelled but the C-terminus has been added to all panels in Figure 1.

Changes to ms:

p.17, Figure legend 1: The protein adopts a helical bundle structure similar to other BCL-2 proteins (RMSD 2.4Å over 141 residues *versus* BCL-XL (PDB 1PQ0); TM-score: 0.83303⁶⁹)

References: A reference to the comparison tool (TM-align) has now been added (ref 69).
Figure 1 modified so C-terminus now labelled.

REVEIEWER 3

1. *Figures 5 and 6 are well presented and contain proper positive controls necessary due to convincingly demonstrate the negative results regarding lack of physical interactions between T. suis A and T. suis B, and the functional apoptotic assays. However, I do believe that the conclusion present on page 7 should present the limitation that these studies were done within a mammalian cell culture system. It is still possible that in its proper in vivo context, T. suis A and T. suis B may interact and/or act like BAX/BAK to promote apoptosis.*

Response:

This is a good point. We have now included the added text to the Discussion to cover this suggestion.

Change to ms:

p.8, added:

One caveat to all of these studies is that they were conducted using a mammalian tissue culture system. Whilst this has been used successfully to dissect the function of BCL-2 family members from other organisms, such as schistosomes¹², it is possible that, in some cases, the native physiological cellular environment might be required to reveal the pro- or anti-apoptotic activity of the proteins.

2. *At the bottom of page 6, the authors describe experiments using synthetic peptides and recombinant proteins to further investigate the potential interaction between T. suis A and B. This data is not presented and should be (perhaps as a supplemental) if the authors deem it worthwhile to include in the text.*

Response:

The reason a figure was not included for this data is that “non-binders” appear to give “noisy” data. Nevertheless, we have now included plots for all of these data in a new Supplementary Figure 2.

Change to ms:

-p.6. Reference to Supplementary Figure 2 included: “...as seen for mammalian BCL-2 proteins (Supplementary Figure 2).”

-New Supplementary Figure 2 and legend added to Supplementary Materials.

3. *The authors when describing the crystal structures at the bottom of page 3 and top of page 4 could make it clearer in the text from that they solve the structure of one of the two T. suis BCL-2 orthologues and that the one they solve is referred to as T. suis A. The second T. suis form is not mentioned until later and the fact that the structure is T. suis A is apparent Figure 2 legend, but I believe that this should be made clearer earlier in the text.*

Response:

This is also a good point. We did attempt to obtain crystals for T. suis B and trialled ~1000 conditions but never got any that could be used. We have now clarified this in the text.

Change to ms:

p.4, the following sentences have been modified to include mention of both T. suis proteins:

Thus, due to the functional importance of this region in *C. elegans*, we attempted to determine the crystal structure of both BCL-2 proteins of *T. suis* (designated as *T. suis* A and *T. suis* B for convenience), a representative species of clade I. We were only able to obtain crystals for *T. suis* A and determined its structure to high resolution (1.6 Å) (Supplementary Table 5), revealing a classic BCL-2-fold consisting of 8 α -helices (Figure 1B).

REVIEWERS' COMMENTS:

Reviewer #1 (Remarks to the Author):

I thank the authors for their careful consideration of my comments and suggestions. The authors have addressed all comments and concerns and clarified all concerns.

Reviewer #3 (Remarks to the Author):

All my concerns have been addressed.